# Effects and Mechanism of Particulate Matter on Tendon Healing Based on Integrated Analysis of DNA Methylation and RNA Sequencing Data in a Rat Model

**DOI:** 10.3390/ijms23158170

**Published:** 2022-07-25

**Authors:** Su-Yel Lee, Min-Hyeok Lee, Seong-Kyeong Jo, In-Ha Yoo, Boler-Erdene Sarankhuu, Hyun-Jin Kim, Yea-Eun Kang, Seong-Eun Lee, Tae-Yeon Kim, Moon-Hyang Park, Choong-Sik Lee, Seung-Yun Han, Ji-Hyun Moon, Ju-Young Jung, Geum-Lan Hong, Nam-Jeong Yoo, Eun-Sang Yoon, Jae-Kyu Choi, Ho-Ryun Won, Ji-Woong Son, Jae-Hwang Song

**Affiliations:** 1Myunggok Medical Research Institute, College of Medicine, Konyang University, Daejeon 35365, Korea; midanf@hanmail.net; 2Division of Pulmonology, Department of Internal Medicine, Konyang University Hospital, Daejeon 35365, Korea; spacetravel@naver.com; 3Department of Orthopedic Surgery, Konyang University Hospital, Daejeon 35365, Korea; joseongkyeong@gmail.com (S.-K.J.); 400852@kyuh.ac.kr (J.-K.C.); 4Department of Medical Science, Asan Medical Institute of Convergence Science and Technology, Asan Medical Center, University of Ulsan College of Medicine, Seoul 05505, Korea; yih2576@naver.com; 5Department of Pharmacology, College of Medicine, Konyang University, Daejeon 35365, Korea; azbolor84@gmail.com; 6Department of Mechanical Engineering, Korea Advanced Institute of Science and Technology, Daejeon 34141, Korea; kim.hyunjin@kaist.ac.kr; 7Research Institute for Medicinal Sciences, College of Medicine, Chungnam National University, Daejeon 35015, Korea; yeeuni220@naver.com (Y.-E.K.); koala0316@hanmail.net (S.-E.L.); 8Division of Endocrinology and Metabolism, Department of Internal Medicine, College of Medicine, Chungnam National University, Daejeon 35015, Korea; 9Bio-Synergy Research Center, Daejeon 34141, Korea; ty.kim1224@guest.kaist.ac.kr; 10Department of Pathology, College of Medicine, Konyang University, Daejeon 35365, Korea; parkmh@hanyang.ac.kr (M.-H.P.); 200635@kyuh.ac.kr (C.-S.L.); 11Department of Anatomy, College of Medicine, Konyang University, Daejeon 35365, Korea; jjzzy@konyang.ac.kr (S.-Y.H.); smcmjh@naver.com (J.-H.M.); 12Department of Veterinary Medicine & Institute of Veterinary Science, Chungnam National University, Daejeon 34134, Korea; jyjung@cnu.ac.kr (J.-Y.J.); ghdrmafks@o.cnu.ac.kr (G.-L.H.); 13College of Medicine, Konyang University, Daejeon 35365, Korea; chaeholove@naver.com (N.-J.Y.); terryyoon0811@gmail.com (E.-S.Y.); 14Department of Otolaryngology-Head and Neck Surgery, College of Medicine, Chungnam National Univesity, Daejeon 35015, Korea; hryun83@gmail.com

**Keywords:** particulate matter, tendon healing, Achilles tendon, rat, DNA methylation, RNA sequencing, CREB signaling

## Abstract

Exposure to particulate matter (PM) has been linked with the severity of various diseases. To date, there is no study on the relationship between PM exposure and tendon healing. Open Achilles tenotomy of 20 rats was performed. The animals were divided into two groups according to exposure to PM: a PM group and a non-PM group. After 6 weeks of PM exposure, the harvest and investigations of lungs, blood samples, and Achilles tendons were performed. Compared to the non-PM group, the white blood cell count and tumor necrosis factor-alpha expression in the PM group were significantly higher. The Achilles tendons in PM group showed significantly increased inflammatory outcomes. A TEM analysis showed reduced collagen fibrils in the PM group. A biomechanical analysis demonstrated that the load to failure value was lower in the PM group. An upregulation of the gene encoding cyclic AMP response element-binding protein (CREB) was detected in the PM group by an integrated analysis of DNA methylation and RNA sequencing data, as confirmed via a Western blot analysis showing significantly elevated levels of phosphorylated CREB. In summary, PM exposure caused a deleterious effect on tendon healing. The molecular data indicate that the action mechanism of PM may be associated with upregulated CREB signaling.

## 1. Introduction

Power generation using fossil fuels (e.g., coal, oil, and natural gas) is a large contributor to the emission of aerosol particles and its precursors [1]. The main emissions from fossil fuel combustion are called primary particulate matter (PM) and secondary PM, including sulfur dioxide (SO_2_) and nitrogen oxides (NO_x_), which are precursors of PM < 2.5 µm in size (PM2.5) [2]. Several studies have provided clear evidence of the growing PM-related health issues worldwide [3]. Specifically, exposure to PM2.5 increases the risk of developing inflammatory lung diseases and lung cancer [3]. PM inhalation can also induce other extrapulmonary diseases via systemic inflammation through cytokine release to the bloodstream [4,5]. In addition, PM is associated with the progression of various diseases through genomic, transcriptomic, and epigenomic alterations [6,7,8]. The effects of PM on the genome have been well-documented in both in vitro and human studies [8]. Therefore, the impact of PM inhalation on human health and the development of strategies for reducing PM exposure have become chief scientific and societal concerns [3]. Importantly, investigating the health effects of PM should not be limited to the lungs, where PM is directly deposited, but should also involve the systemic or peripheral organs, such as the musculoskeletal system. However, currently available data on PM-induced inflammatory responses and epigenetic effects were mainly obtained from studies on cancer and respiratory, cardiovascular, and neurodegenerative diseases [3,6,7,8]. Recently, several studies have demonstrated the effects of PM and toxin inhalation on musculoskeletal diseases. Peng et al. [9] suggested that osteoarthritis severity is promoted by PM exposure via systemic inflammatory mechanisms [9]. The detrimental effects of cigarette smoking on tendon healing and other musculoskeletal disorders have also been reported [10,11]. Additionally, Lehner et al. [5] demonstrated that intranasal allergen instillation in sensitized mice can induce systemic inflammation and tendinopathy.

Hence, we speculated that exposure to PM2.5 may also influence tendon healing. Tendon injuries, such as Achilles tendon rupture (ATR) and rotator cuff tears, are common orthopedic conditions that cause significant morbidity [12,13]. Furthermore, the incidence of tendon injuries has increased with greater societal participation in recreational and competitive sports activities [14]. However, to our knowledge, no experimental study has been conducted to determine the relationship between tendon healing and PM exposure. We hypothesized that PM exposure may substantially influence tendon healing. Here, we performed multi-pronged histological [15], ultrastructure morphological [16], and biomechanical [17] investigations using well-established methods to reveal the effects of PM in an ATR rat model. Additionally, we performed the combined analysis of DNA methylation and RNA sequencing (RNA-seq) data to elucidate the mechanism underlying the PM-induced effects on tendon healing.

## 2. Results

### 2.1. PM Exposure Induced Lung Inflammation

Gross images of the lungs from the PM group showed multiple darkly spotted lesions (Figure 1). A subsequent histological analysis revealed inflammation of the alveolar walls and increased numbers of lymphocytes and histiocytes. The interalveolar septa were also thickened after inflammatory cell infiltration.

### 2.2. PM Exposure Induced Systemic Inflammation Demonstrated by Laboratory Blood Test and ELISA (Enzyme-Linked Immunosorbent Assay)

The complete blood count (CBC) showed that the PM group had a significantly elevated white blood cell (WBC) count compared to the non-PM group (7.2 ± 1.6 vs. 4.5 ± 1.9 × 10^9^/L, *p* = 0.04). However, no significant difference was observed between the segmented neutrophil counts of the two groups (1.1 ± 0.1 vs. 0.8 ± 0.3 × 10^9^/L, *p* = 0.171). Notably, ELISA demonstrated that tumor necrosis factor-alpha (TNF-α) expression was significantly higher in the PM group than the non-PM group (*p* = 0.004).

### 2.3. PM Exposure Had Detrimental Histological Effects on Tendon Healing in Rats with ATR

The modified Bonar score was significantly higher in the PM group than the non-PM group (17.5 ± 0.9 vs. 9.0 ± 3.0, *p* = 0.032) (Figure 2). Based on the histological characteristics, the Bonar scores for cell morphology (*p* = 0.007), collagen fiber arrangement (*p* = 0.007), and ground substance (*p* = 0.016) were increased in the PM group. Furthermore, the Bonar scores for cellularity (*p* = 0.116), vascularity (*p* = 0.230), calcification (*p* = 0.116), and adipocytes (*p* = 0.519) were higher in the PM group than the non-PM group (not statistically significant).

### 2.4. Transmission Electron Microscopy (TEM) Analysis Revealed Decreased Mean Diameter of Collagen Fibrils in PM Group

Cross-sectional TEM images of the collagen fibrils in both groups were obtained (Figure 3). The distributions of the collagen fibril diameters within each group were not uniform (Figure 3B). The mean diameter of the collagen fibrils was significantly lower in the PM group than the non-PM group (50.9 ± 6.6 vs. 52.4 ± 8.7 nm, *p* = 0.02) (Figure 3C).

### 2.5. The Load to Failure of the PM Group Was Significantly Decreased Compared to the Non-PM Group

The gross findings of both groups demonstrated the continuity of the Achilles tendons (Figure 4). The load to failure of the PM group was significantly decreased compared to the non-PM group (36.0 ± 9.0 vs. 47.3 ± 4.1 N, *p* = 0.04). The other biomechanical outcomes, including the cross-sectional area (CSA) (11.3 ± 1.5 vs. 11.7 ± 3.9 mm^2^, *p* = 0.429), stiffness (14.2 ± 9.4 vs. 15.0 ± 8.3 N/mm, *p* = 0.452), and stress (3.3 ± 1.2 vs. 4.7 ± 2.8 N/mm^2^, *p* = 0.189), were also reduced in the PM group (not significant).

### 2.6. Integrated Analysis of DNA Methylation and RNA-Seq Data Implied That PM May Have a Considerable Effect on Tendon Healing through cAMP Response Element-Binding Protein 1 (CREB1) Upregulation

After normalization, 389,347 probes were retained for further analysis, and 28,046 differentially methylated probes (7.20%) were identified. Among these, 23,844 (6.12%) were hypomethylated and 4202 (1.07%) were hypermethylated. In addition, 4703 (30.10%) and 573 (3.52%) genes were found to be hypomethylated and hypermethylated at the gene level, respectively.

An RNA-seq analysis revealed the significant differentially expressed genes (DEGs) between the PM and non-PM groups based on the *P*-values and fold change (FC) values. A Kyoto Encyclopedia of Genes and Genome (KEGG) analysis and an ingenuity pathway analysis (IPA) demonstrated that various canonical pathways were significantly associated with the effects of PM exposure on tendon healing. Specifically, DEGs in pathways related to cancer, protein processing in the endoplasmic reticulum, cyclic adenosine monophosphate (cAMP) signaling, calcium signaling, and phosphoinositide 3-kinase (PI3K)-protein kinase B (Akt) signaling were upregulated in the PM group, whereas coronavirus-disease- and ribosome-related pathways were downregulated (Figure 5A). Additionally, the IPA revealed that the cAMP response element-binding protein (CREB), corticotrophin-hormone-releasing, and cAMP signaling pathway proteins were significantly upregulated in the PM group (Figure 5B and Appendix A, available online). Among these, the CREB signaling pathway had the highest Z-score. Moreover, a heatmap analysis exhibited that the PM group showed significantly increased expression of various genes related CREB signaling (Figure 5C) and cAMP signaling (Figure 5D).

A combined analysis of the DNA methylation and RNA-seq data revealed that 67 genes exhibited hypomethylated/upregulated expression, whereas 9 genes had hypermethylated/downregulated expression (Figure 5E,F). To determine the potential biological function of the hypomethylated/upregulated genes, we performed a KEGG pathway analysis and discovered that these genes were significantly enriched in several pathways, including gap junction formation, melanogenesis, and PI3K-Akt signaling (Figure 5G).

Taken together, these results imply that PM may have a considerable effect on tendon healing through CREB1 upregulation via specific mechanisms such as cAMP and PI3K-Akt signaling.

### 2.7. p-CREB Expression Was Significantly Elevated in the PM Group Compared to the Non-PM Group in the Western Blot Analysis

A quantitative analysis of the proteins related to the cAMP and PI3K-Akt signaling pathways was performed to validate the molecular data (Figure 6). Additionally, the expression of proteins involved in tendon regeneration was investigated. Compared to the non-PM group, the cAMP (*p* = 0.021) and p-Akt (*p* = 0.007) expression levels in the PM group were also significantly higher than those in the non-PM group. In addition, the p-CREB expression was significantly elevated in the PM group compared to the non-PM group (*p* = 0.031). Compared to the non-PM group, the serum TNF-α level was significantly higher in the PM group (*p* = 0.009), whereas collagen type I (COL1) expression (*p* = 0.004) was significantly lower in the PM group.

## 3. Discussion

The findings in this study support our hypothesis that tendon healing is substantially affected by the PM exposure. Notably, we discovered that PM exposure had detrimental histological and biomechanical effects on tendon healing in rats with ATR. Our findings also suggest that the upregulated CREB signaling pathway is closely related to the effect of PM.

The incidence of acute ATR has increased in the past decades and was reported at 0.27 per 100 person-years [18]. The optimum treatment still remains controversial, and operative and conservative treatments have distinct advantages and disadvantages [12]. Regardless of treatment method, rapid tendon healing and early rehabilitation after ATR are essential for successful clinical outcomes [12]. Recently, there is increasing interest in factors affecting tendon healing after ATR; however, the majority of previous studies mainly focused on comparing the treatment methods instead of investigating environmental factors [12,19].

In acute tendon injuries, the acute inflammatory, proliferative, and remodeling phases define the regeneration process [20]. Acute and chronic systemic inflammation and the immune conditions of individuals play important roles during these phases [20,21]. The prolonged state of low-grade inflammation in tendinopathy may be a risk factor for a ‘failed healing response’ following acute tendon insult, predisposing affected individuals to disrupted healing [20]. A recent study reported that intranasal allergen instillation in sensitized mice induced systemic inflammation and tendinopathy in a mouse model [5]. Thus, we hypothesized that exposure to and inhalation of certain environmental factors, such as PM, may detrimentally affect tendon healing via systemic conditions. PM inhalation can induce the development of extrapulmonary diseases via systemic inflammation through cytokine release into the bloodstream [4,5]. In addition, PM can promote the progression of many diseases by inducing epigenetic alterations [6,7,8]. Hence, we investigated the outcome of tendon healing through systemic inflammation and epigenetic changes after PM exposure using an ATR rat model [22].

Here, PM2.5 was used for the PM exposure experiments due to its small size, which may be capable of reaching the end of the respiratory tract via airflow and subsequently accumulate by diffusion, thereby damaging other parts of the body through air exchange in the lungs [23]. To ensure the sufficient PM2.5 concentration in the closed chambers, real-time monitoring using a PM sensor was performed to maintain the concentration level above 300 µg/m^3^, following the current World Health Organization guidelines stating that annual average concentrations of PM2.5 should not exceed 5 µg/m^3^, while 24 h average exposures should not exceed 15 µg/m^3^. Furthermore, to check the adequate flow of PM in the closed chamber, the analysis of PM circulation and streamlines in the chamber was performed by computer simulation.

The lung and systemic inflammation caused by PM exposure were examined via a histological analysis and blood tests, respectively. The histological analysis of the lung tissues revealed the inflammation of the alveolar walls and an increased number of macrophages in PM-exposed rats. Furthermore, the interalveolar septa were thickened after inflammatory cell infiltration. Compared to the non-PM group, lymphocytic vasculitis and foamy histiocytes were more frequently observed in the PM group [24]. These results demonstrate that PM infiltrated and affected the lungs in the PM group using our method. A significantly elevated WBC count and higher TNF-α expression were also observed in the PM group. As a pro-inflammatory cytokine, TNF-α was measured via serum ELISA to investigate the systemic inflammation caused by lung injury [25]. These data indicate that PM induced systemic inflammation via cytokine release to the bloodstream as well as pulmonary inflammation. Interestingly, PM treatment reportedly promotes inflammatory infiltration in the lungs and increases the serum levels of TNF-α and interleukin 1 beta (IL-1β) [25].

A macroscopic analysis 6 weeks post-operation revealed that the tendons in both groups presented as hard, thickened, and pink structures with continuity, and no gap was evident in the tendon callus of either group. The histopathological changes associated with tendon injury include cellular changes, collagen disruption, and enhanced glycosaminoglycan levels [15,26,27]. The Achilles tendon in the PM group exhibited significantly more inflammatory outcomes than that in the non-PM group. Among the histological parameters, cell morphology, collagen fiber arrangement, and the accumulation of ground substances showed significant differences between the two groups. In the PM group, round and enlarged nuclei with visible cytoplasm were evident, and a marked disarrangement of collagen with abundant ground substances in the extracellular matrix was observed. Additionally, intratendinous calcification and ossification (chondroid metaplasia and bone formation) were more frequently observed in the PM group than the non-PM group. These outcomes suggest that a prolonged maturation of fibroblasts into fibrocytes and an altered collagen arrangement, which are indicative of delayed histological recovery, occurred in the PM group [19].

Since cross-sections show the diameter and distribution of collagen fibrils, TEM analysis is a useful method for evaluating the tendon ultrastructure [16]. Here, a TEM analysis revealed the decreased mean diameter of collagen fibrils in the PM group. A biomechanical assessment also showed a significantly lower load to failure value in the PM group compared to the non-PM group. We speculate that PM-induced systemic inflammation and epigenetic changes may have delayed the restoration and maturation of the collagen fibers, leading to poor biomechanical results. Previous studies reported that inferior biomechanical recovery ultimately results in delayed functional recovery during tendon healing [28]. Clinically, the decreased diameter of collagen fibrils might induce decreased motor power of ankle plantarflexion, which might substantially affect the daily life and sports activity of injured patients.

To elucidate the action mechanism of PM, we examined the PM-induced epigenetic changes on tendon healing. Next-generation sequencing technologies have revolutionized genome sequencing, producing large quantities of genetic data [29]. As one of the most studied epigenetic changes in human cells, the changes in DNA methylation patterns are crucial in the development of various diseases [29]. Additionally, RNA-seq is an advanced technique for analyzing and quantifying transcriptomes. In this study, a combined analysis of DNA methylation and RNA-seq analyses revealed the significantly increased expression of PI3K-Akt and *CREB1* in PM-exposed rats. A further analysis showed that the CREB and cAMP signaling pathways were also significantly upregulated in the PM group. Since several protein kinases can phosphorylate CREB, it is a convergent target for multiple intracellular signaling cascades [30]. CREB contains multiple phosphorylation sites for different signaling pathways, including PI3K-Akt and cAMP [30]. A Western blot analysis validated these results, revealing that the expression of proteins related to the PI3K-Akt, cAMP-PKA, and CREB pathways were upregulated. Overall, these findings imply that PM may influence tendon healing via *CREB1* upregulation through the cAMP and PI3K-Akt signaling pathways.

Alveolar macrophages and different types of epithelial cells constitute the primary targets of inhaled lung toxicants and are particularly important in the induction of inflammatory responses to PM exposure [31]. Macrophages release various inflammatory cytokines upon particle exposure, including TNF-α, IL-6, and IL-1β [25,31]. The increased number of histiocytes in the lungs and the elevated TNF-α level in the blood after PM inhalation demonstrate that systemic inflammation was successfully induced in our in vivo model. TNF-α secreted into the blood promotes systemic inflammation and organ dysfunction by affecting various cellular signaling pathways [32,33]. By binding to cell receptors, TNF-α activates downstream pathways (e.g., PI3K-Akt, cAMP, and p38/MAPK) and induces different biological effects depending on the cell type [32,33]. Tenocytes also have receptors for TNF-α, and in vitro stimulation of tenocytes by TNF-α can aggravate inflammation in tendinopathy [34]. Hence, we presume that PM-induced circulatory TNF-α secretion activated the PI3K-Akt and cAMP signaling pathways in the Achilles tendon, thereby stimulating the associated CREB pathway.

CREB reportedly induces the transcription of immune-related genes that possess CRE elements, including TNF-α and IL-6, which are well-known pro-inflammatory cytokines that stimulate inflammation in tendon healing [35,36,37]. Thus, the sustained increase in TNF-α can induce inadequate inflammation in the tendon [38]. John et al. [39] reported that TNF-α-treated tenocytes exhibit increased TNF-α and matrix metalloproteinase 1 levels and decreased COL1 levels [39]. Several studies reported that TNF-α also inhibits collagen expression and promotes collagenase expression in fibroblasts [40,41]. In this study, elevated TNF-α expression and inhibited COL1 expression were detected in the PM group.

To our knowledge, this is the first study to investigate the effects of PM exposure on tendon healing. Our multi-pronged histological [15], ultrastructure morphological [16], and biomechanical [17] approaches using well-established methods successfully revealed the effects of PM in an ATR rat model. The use of PM2.5, which is capable of reaching the end of the respiratory tract with airflow, and an analysis of PM circulation in the chamber using a computer simulation further improved the methodology. Additionally, the combined analysis of DNA methylation and RNA-seq data to elucidate the mechanism underlying the PM-induced effects on tendon healing revealed novel targets, including the CREB pathway.

## 4. Materials and Methods

### 4.1. Animals and Experimental Outline

All experiments involving animals were approved by the Institutional Animal Care and Use Committee of Konyang University (approval number: P-21-04-A-01). The protocol followed the tenets of the “Guide for the Care and Use of Laboratory Animals” (National Institutes of Health (NIH), Bethesda, MD, USA) [42]. The schedule for the in vivo experiments is shown in Figure 7. Briefly, 20 sexually mature male Sprague–Dawley rats (250–300 g, 8 weeks old) purchased from Samtako (Osan, Korea) were subjected to open Achilles tenotomy [22]. Then, animals were randomly assigned to one of two groups according to exposure to PM: a PM exposure group (PM group, n = 10) and a control group (non-PM group, n = 10). The PM group was exposed to PM2.5 thrice weekly for 6 weeks. On day 43, the rats were anesthetized using 3% isoflurane (70 N_2_O:30 O_2_). Before euthanasia, blood samples (10 mL; n = 5 per group) for laboratory tests and ELISA were collected from the abdominal aorta, and the lungs and tendons were harvested, following saline intracardiac perfusion, for further examination [43]. A histological analysis of the lungs was performed (n = 3 each), and the collected tendons were analyzed via histological examination and TEM (n = 3 each), biomechanical assessment (n = 4 each), DNA methylation microarray and RNA-seq (n = 3 each), and Western blot analysis (n = 3 each).

### 4.2. Open Achilles Tenotomy

Before the operation, all rats were anesthetized using 3% isoflurane (70 N_2_O:30 O_2_). The right leg of each rat was shaved and sterilized using alcohol. A 15 mm posterior longitudinal incision was made using a No. 11 scalpel blade, and the fascia was dissected to expose the Achilles and plantaris tendons. The tendons were completely transected perpendicular to the collagen fibers proximal (5 mm) to the calcaneal insertion [22]. The skin was closed by simple sutures using Nylon 4-0 (Ethicon, Somerville, NJ, USA). Cast immobilization was not applied to the legs. All animals were permitted to move within their cages and had ad libitum access to food and water.

### 4.3. PM Collection and Exposure

The PM2.5 sample was collected from an experimental coal-fired power plant at the Korea Institute of Energy Research (Daejeon, Korea) [44]. Particles < 10 µm were collected via treatment with multi-cyclones. A size analysis using a laser diffraction spectrometer revealed that 90% of the particles were <10.49 µm, whereas 50% were <2.14 µm. The PM group was exposed to PM2.5 for 30 min daily [45] and thrice weekly for 6 weeks in closed chamber systems (Figure 8). An InnoSpire Essence Compressor Nebulizer System (Philips Respironics, Murrysville, PA, USA), with a maximum pressure of 371 kPa and a maximum flow rate of 8.0 L/min, was used to scatter 120 mg of PM in the closed chamber. The PM concentration was closely monitored in real time using an FM-322 Laser PM Sensor (CAS, Seoul, Korea). During monitoring, an additional 240 mg of PM (120 mg every 10 min) was scattered in the closed chamber to maintain the PM2.5 level above 300 µg/m^3^.

### 4.4. Flow Simulation in the Chamber

To ensure the adequate exposure of the animals to PM, the PM circulation and streamlines in the chamber were analyzed using computer simulations (Figure 8). The airflow in the experiment can be approximated as a Newtonian compressible fluid obeying the ideal gas law because the Mach number and temperature variation were low. For the chamber defined as the spatial domain, Ω, with boundary Γ, the airflow was calculated using time-dependent compressible Navier–Stokes equations:$$U_{,t}+F_{I,i}^{adv}-F_{I,i}^{diff}=f\;in\;\Omega$$
with the variables given by $U=\left(\begin{array}{c} \begin{array}{c} U_{1}\\ U_{2}\\ U_{3} \end{array}\\ U_{4}\\ U_{5} \end{array}\right)=\rho \left(\begin{array}{c} \begin{array}{c} 1\\ u_{1}\\ u_{2} \end{array}\\ u_{3}\\ e_{tot} \end{array}\right)$ and the advective and diffusive fluxes defined as
$$F_{i}^{adv}=u_{i}U+P\left(\begin{array}{c} \begin{array}{c} 0\\ \delta_{1i}\\ \delta_{2i} \end{array}\\ \delta_{3i}\\ u_{i} \end{array}\right),\;F_{i}^{diff}=\rho \left(\begin{array}{c} \begin{array}{c} 0\\ \tau_{1i}\\ \tau_{2i} \end{array}\\ \tau_{3i}\\ \tau_{ij}u_{j}-q_{i} \end{array}\right).\;$$

These equations were closed through the introduction of the constitutive relationships given by
$$\tau_{ij}=2\;\mu \;\left(S_{ij}\;\left(u\right)-\frac{1}{3}\;S_{kk}\left(u\right)\delta_{ij}\right),\;S_{ij}\left(u\right)=\frac{u_{i,j}+u_{j,i}}{2}.\;$$
$$\;q_{i}=-k\;T_{,i},\;e_{tot}=e+\frac{u_{i}u_{i}}{2},\;e=c_{v}\;T,\;P=\rho RT$$
where $u,\;P,\;T,\;e_{tot}$, ρ, μ, k, R, and f represent the fluid velocity, pressure, temperature, total energy, density, viscosity, molecular conductivity, individual gas constant, and external force at position x and time *t*, respectively [46,47,48]. The fluid velocity was set to zero, the temperature was set to room temperature, and the pressure was set to atmospheric pressure for the initial condition. The boundary, Γ, of the chamber can be split into two partitions, such that $\Gamma =\Gamma_{in}\cup \;\Gamma_{wall}$ and $\Gamma_{in}\cap \;\Gamma_{wall}=\emptyset .$ A constant airflow of 9.4 L/min was applied for the first 20 s at the inlet $\Gamma_{in}\;$, and zero flow was applied for the remaining 9 min and 40 s. No slip boundary conditions were set for the chamber wall $\Gamma_{wall}$, and the temperature at the boundary Γ was set at room temperature. The Navier–Stokes equations were formulated using a stabilized Galerkin finite element method and solved using the matrix-free generalized minimal residual (MF-GMRES) solution with a block diagonal preconditioner developed via element-by-element GMRES. The second-order generalized α method was utilized for the time integration of the Navier–Stokes equations [49,50]. The air was modeled using the following parameters: individual gas constant, R, 287.05 J/kg/K; dynamic viscosity, μ, 1.8 $\times 10^{-5}$ kg/m/s; molecular conductivity, k, 0.025 W/m/K; Prandtl’s number, 0.72; and room temperature, 20 °C [51].

### 4.5. Laboratory Blood Test and ELISA

Laboratory tests of whole-blood samples were performed to determine the CBC, specifically the WBC and segmented neutrophil counts, using an ADVIA 2120i Hematology System (Siemens Healthcare Diagnostics, Deerfield, IL, USA). The serum levels of TNF-α were measured using rat TNF-α (#438207, BioLegend, San Diego, CA, USA) and ELISA kits, following the manufacturer’s instructions [25].

### 4.6. Histological Analysis

For histological examinations, the rat lungs and Achilles tendons were placed in paraffin blocks and then cut using a tissue microtome (RM2255, Leica, Heidelberger, Germany). Serial paraffin sections of the lung tissues (5 μm thick) were used for hematoxylin and eosin (H&E) staining [9]. A tissue block (5 × 15 mm) consisting of Achilles tendon and peritendinous tissue between the calcaneus and musculotendinous junctions was obtained, serially sectioned (5 μm thickness), and stained with H&E, Masson’s trichrome, and Alcian blue [22]. The sections were examined using a digital camera connected to a DM4 light microscope (Leica Microsystems, Weltzlar, Germany). The lung specimens were evaluated for interalveolar septum and inflammatory cell infiltration, whereas the tendon specimens were assessed using the Bonar score, following the standardized recommendations from a previous study [15]. After an initial analysis of the entire tendon (100× magnification), the region with the greatest alteration in cell morphology was identified and characterized using the following characteristics: cell morphology, collagen fiber arrangement, the accumulation of ground substance, cellularity, and vascularization. Each characteristic was graded on a scale ranging from 0 (normal) to 3 (advanced changes). The total Bonar score per specimen was calculated using the sum of the characteristic grades, with an additional 2.5 points for those with intratendinous calcification and adipocytes. Hence, a tendon exhibiting the most severe pathology and a tendon with no observable pathology would have scores of 20 and 0, respectively [15].

### 4.7. TEM Analysis

Tissue samples from Achilles tendons were fixed with 2.5% glutaraldehyde and washed with 0.1 M phosphate buffer (pH 7.4) at 4 °C. The samples were post-fixed with 1% osmium tetroxide (Electron Microscopy Sciences, Hatfield, PA, USA) in 0.1 M phosphate buffer solution for 1 h at 4 °C, dehydrated using gradient ethanol solutions (50%, 70%, 80%, 90%, 95%, and 100%), and placed in propylene oxide. The samples were progressively embedded in 1:1, 1:2, and 0:2 ratios of propylene oxide and EMbed-812 resins, followed by polymerization at 60 °C for 48 h. The tissues were sectioned (70 nm thickness) with diamond knives using an RMC Ultramicrotome (RMC Boeckeler, Tucson, AZ, USA) and mounted on 200 mesh copper grids. The sections were post-stained using 2% uranyl acetate and 1% lead citrate and observed using an HT7700 TEM (Hitachi, Tokyo, Japan) at 80 kV. Images of the collagen fibers were obtained and used to measure the fibril diameter distribution and mean diameter [52]. Approximately 100 collagen fibrils were measured per sample (n = 3 each) using ImageJ software version 1.53 (NIH).

### 4.8. Biomechanical Assessment

The biomechanical properties of the tendons from each group were determined using the established methodology for the biomechanical assessment of ATR in rats [22]. Each tendon was harvested with the attached calcaneal bone and musculotendinous portion. Assuming an elliptical tendon cross-section, the CSA at the midportion of the tendon callus was calculated based on two perpendicular axes of thickness (major axis a and minor axis b) using the formula: CSA = π × a × b/4 [53]. Thickness was measured using an SD500-150PRO Digimatic Caliper (Sincon, Shanghai, China). The tendon with the attached calcaneal bone was wrapped in gauze soaked in saline prior to mechanical testing. The Achilles tendon was securely placed in serrated metal grips and mounted on an EZ-SX Texture Analyzer (500 N; Shimadzu, Kyoto, Japan). The calcaneus and musculotendinous junction were fixed with clamps at the distal and proximal ends of the specimen, respectively. The tendon was pulled perpendicular to the floor at a constant speed (0.1 mm/s) until failure, and the load to failure (N), stiffness (N/mm), and stress (N/mm^2^) values were recorded [19]. Stiffness was plotted by measuring the linear slope in the load–displacement curve graph prior to achieving the maximum load to failure [54]. Stress was calculated as the load to failure divided by the CSA [55]. The parameter calculation data were collected using Trapezium 2 software version 2.05 (Shimadzu).

### 4.9. DNA Methylation Microarray

A DNA methylation microarray was performed by Genomictree (Daejeon, Korea) using the tendon tissues (n = 3 per group) and a custom-designed Agilent-based microarray platform with 2 × 400 K probes per slide (popular_2X400K_chip; Agilent Design ID: 086791; Agilent Technologies, Santa Clara, CA, USA). We successfully designed 60-mer probes for 389,347 measurements using eArray software (Agilent Technologies. http://earray.chem.agilent.com/earray/), following the standard probe design criteria. Genomic DNA was extracted from the tendons using a DNA mini kit (Qiagen, Hilden, Germany), checked for purity using a Nanodrop Spectrophotometer (Thermo Fisher Scientific, Waltham, MA, USA), and then fragmented (1.0 μg) via sonication (10 s on/off, 24×, 40%). The fragmented DNA was incubated with MBD2bt for enrichment and purified using nickel-coated magnetic beads. The enriched DNA fragments were amplified using a Whole Genome Amplification Kit (Sigma-Aldrich, St. Louis, MO, USA). After labeling with Cy3 and Cy5 for the control and PM groups, respectively, the DNA was hybridized to the chip using an Agilent ChiP-on-chip/aCGH Hybridization Kit. The images were scanned and quantified using an Agilent DNA Microarray Scanner and Feature Extraction Software 11.5.1.1 (Santa Clara, CA, USA), respectively. The hypermethylated/hypomethylated probes were selected via the normalization of array data following the LOWESS intensity-dependent algorithm in GeneSpring version 7.2 (Silicon Genetics, Redwood City, CA, USA) using Student’s *t*-test (*p*-value < 0.05). Two adjacent probes with FC values ≥ 2.0 or ≤0.5 were defined as differentially methylated genes. The GEO submission number is GSE200709.

### 4.10. RNA-Seq Analysis

The healed portion of the Achilles tendon was collected en bloc as 1 × 0.5 cm pieces from both groups. Total RNA was extracted from the rat tendon samples (100 mg) using the Trizol reagent (Ambion, Austin, TX, USA), according to the manufacturer’s instructions. To perform RNA quality control, RNA was quantified spectrophotometrically using a NanoDrop 2000 spectrophotometer (Thermo Fisher Scientific, Waltham, MA, USA). The generation of cDNA and RNA-Seq libraries was conducted by a TruSeq Stranded mRNA LT Sample Prep kit (Illumina at San Diego, CA, USA) following the TruSeq Stranded mRNA Sample Preparation Guide, Part # 15031047 Rev. E and sequenced by the paired-end sequencing method using an Illumina NovaSeq 6000 system sequencer (Macrogen, Seoul, Korea). The RNA sequence data quality was checked using FastQC v0.11.7 (FastQC. http://www.bioinformatics.babraham.ac.uk/projects/fastqc/), and the data were trimmed using Trimmomatic 0.38 to remove low-quality bases and adaptor sequences. The preprocessed reads were aligned to a reference genome (Rnor_6.0) using the HiSAT2 version 2.1.0 alignment program (HISAT2. http://daehwankimlab.github.io/hisat2/). After mapping reads, String Tie version 2.1.3b (String Tie. https://ccb.jhu.edu/software/stringtie/) was used for transcript assembly, and the expression profiles were calculated as fragments per kilobase of transcript per million mapped reads (FPKM). Heatmaps were generated using PermutMatrix [56] version 1.9.3 (LIRMM. http://www.lirmm.fr/~caraux/PermutMatrix/). Genes with *p*-values < 0.05 and FC values ≥ 1.2 between the PM and non-PM groups were defined as DEGs. Functional characterization was performed using KEGG and IPA (Ingenuity Systems, Redwood City, CA, USA) to identify pathways with significant *p*-values and Z-scores [57]. The GEO submission number is GSE200463.

### 4.11. Combined Analysis of DNA Methylation and RNA-Seq Data

A Venn diagram of the significantly expressed genes in the DNA methylation and RNA-seq data was generated to identify the hypomethylated/upregulated and hypermethylated/downregulated genes using an online tool (Bioinformatics & Systems Biology, Gent, Belgium. https://bioinformatics.psb.ugent.be/webtools/Venn/). The identified genes were subjected to a KEGG pathway analysis using GSEApy (gseapy. https://pypi.org/project/gseapy/), and pathways with *p*-values < 0.05 were considered significant.

### 4.12. Western Blot Analysis

The protein expression levels of cAMP, phosphorylated Akt, phosphorylated CREB, TNF-α, COL1, and COL3 were measured via Western blotting. Proteins were extracted from the Achilles tendon tissues by homogenization using Pro-Prep Protein Extraction Solution (iNtRON Biotechnology, Seongnam, Korea). Homogenates were centrifuged at 13,000× g for 20 min at 4 °C. The protein concentration was measured using BCA Protein Assay Reagent (Pierce Biotechnology, Rockford, IL, USA). After gel electrophoresis and hybridization, the proteins were incubated with the following primary antibodies: 1:1000 anti-cAMP (500-9534; Abbomax, San Jose, CA, USA), anti-pAKT (sc-377556; Santa Cruz Biotechnology, Santa Cruz, CA, USA), anti-pCREB (ab32096; Abcam, Waltham, MA, USA), anti-TNF-α (sc-52746; Santa Cruz Biotechnology), anti-COL1 (MA1-26771; Invitrogen, Carlsbad, CA, USA), anti-COL3 (PA5-27828; Invitrogen), 1:2000 goat anti-rabbit IgG (H + L), horseradish peroxidase (HRP, 31463; Invitrogen), 1:2000 goat anti-mouse IgG (H + L), and HRP (31430; Invitrogen). Signals were detected using an Immobilon Chemiluminescence Kit (Merck, San Jose, CA, USA) and then quantified using ImageJ software version 1.53 (NIH). The band intensity per protein was normalized using β-actin as a control.

### 4.13. Statistical Analysis

The data were analyzed using SPSS version 22.0 (IBM Corp., Armonk, NY, USA). The Shapiro–Wilk test was used to assess data normality, whereas Student’s *t*-test was performed to compare the experimental groups [19,58]. Data are presented as means ± SD, with the significance level set at *p* < 0.05 [19].

## 5. Conclusions

In summary, PM exposure induced detrimental histological, ultrastructural, and biomechanical effects on the tendon healing of rats with ATR. Notably, the CREB signaling pathway may be associated with the action mechanism underlying the PM-induced effects on tendon healing and should be further investigated.

Moreover, tendon injury patients may have an increased risk of developing complications during healing when exposed to high levels of PM. Hence, specific precautions should be adopted.

### Limitations and Future Directions

However, several research gaps must be addressed in future studies. First, the exact action mechanism of PM on tendon healing should be simultaneously demonstrated using in vitro and in vivo studies. Second, epigenetic regulation by microRNAs, exosomes, and other factors on tendon healing after PM exposure should be investigated. Third, since our experiment duration was short, studies with longer time periods are required to reveal the long-term effects of PM exposure on tendon healing. Fourth, the effects of PM exposure on the models without injuries were not reported. We performed a preliminary in vivo study to estimate the effects of PM exposure on the models without injuries. However, there was no significant difference regarding the histologic study between the non-PM group and the PM group among the rat Achilles model without injuries.

Overall, additional in vitro and in vivo studies with longer durations are essential to validate the present findings. Moreover, further in vivo study investigating the effects of PM exposure on the models without injuries by multi-pronged approaches is needed.

## Figures and Tables

**Figure 1 ijms-23-08170-f001:**
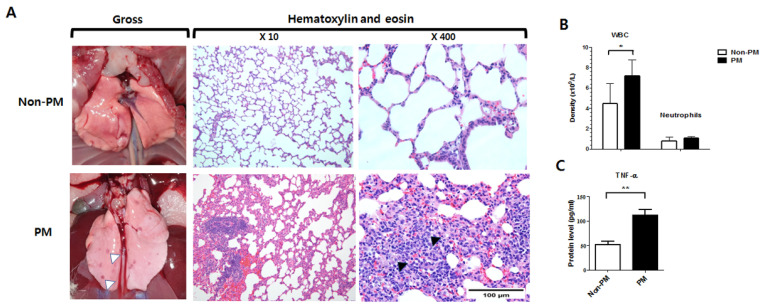
Evaluation of lung and systemic inflammation in particulate matter (PM)-exposed and control (non-PM) rats 6 weeks post-operation. (**A**) Gross and microscopic images of the lungs. Sections were stained with hematoxylin and eosin and observed under 10× and 400× magnification. The microscopic images revealed thickened interalveolar septa with increased numbers of lymphocytes and histiocytes. White arrow heads indicate the darkly spotted lesions in lungs of the PM group. Black arrow heads indicate the histiocytes in alveolar walls of the PM group. (**B**) White blood cell (WBC) and neutrophil densities in whole blood. (**C**) Serum levels of tumor necrosis factor-alpha (TNF-α). Data are presented as means ± SD. * *p* < 0.05, ** *p* < 0.005.

**Figure 2 ijms-23-08170-f002:**
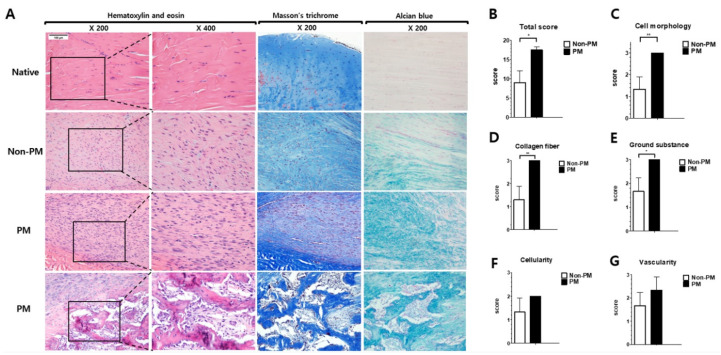
(**A**) Microscopic images of the Achilles tendons from native, control (non-PM), and PM-exposed rats 6 weeks post-operation. Additional images showing ossification (chondroid metaplasia in the marginal area and bone formation in the center) in the PM group are in the last row. Histological specimens were evaluated based on the modified Bonar score. The region with the greatest alteration in cell morphology was identified and evaluated. Images at 200× and 400× magnification of hematoxylin and eosin stain and 200× magnification of Masson’s trichrome stain and Alcian blue stain. Histological results showing the (**B**) total score, (**C**) cell morphology, (**D**) collagen fiber arrangement, (**E**) ground substance, (**F**) cellularity, and (**G**) vascularity of non-PM and PM groups. Data are presented as means ± SD. * *p* < 0.05, ** *p* < 0.005.

**Figure 3 ijms-23-08170-f003:**
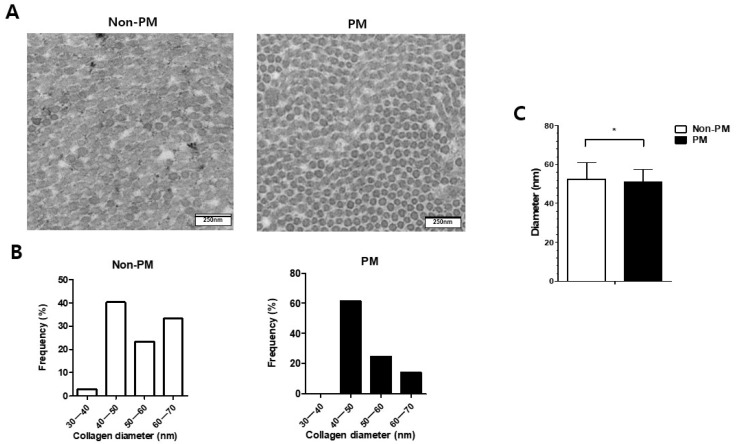
(**A**) Transmission electron microscopy (TEM) images of the collagen fibrils, (**B**) frequency of collagen fibril diameters, and (**C**) mean diameter of collagen fibrils in the Achilles tendons of control (non-PM) and PM-exposed rats 6 weeks post-operation. * *p* < 0.05.

**Figure 4 ijms-23-08170-f004:**
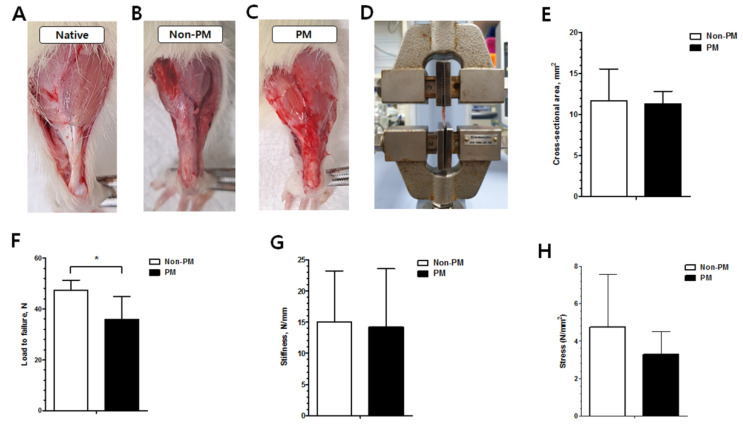
(**A**–**C**) Gross images of the tendons in native, control (non-PM), and PM-exposed rats 6 weeks post-operation. (**D**) Biomechanical testing machine. (**E**) Cross-sectional area, (**F**) load to failure, (**G**) stiffness, and (**H**) stress outcomes. * *p* < 0.05.

**Figure 5 ijms-23-08170-f005:**
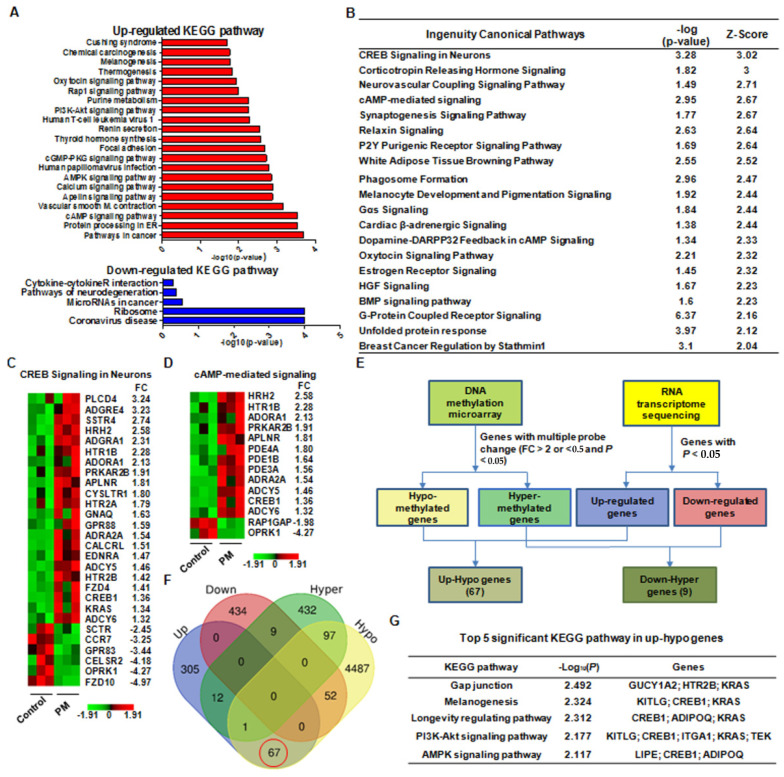
(**A**) Kyoto Encyclopedia of Genes and Genome (KEGG) pathways associated with the significantly upregulated and downregulated differentially expressed genes (DEGs) between particulate matter (PM)-exposed and control (non-PM) rats (*p* < 0.05). (**B**) Canonical pathways with significant Z-scores (>2) identified using ingenuity pathway analysis. Heatmaps showing the DEGs in the (**C**) cAMP response element-binding protein (CREB) and (**D**) cAMP signaling pathways. (**E**) Schematic diagram showing the methods and criteria for gene analysis. (**F**) Venn diagram showing the combined DNA methylation and transcriptome data. The red circle indicates 67 genes exhibiting hypomethylated/upregulated expression. (**G**) The top five KEGG pathways associated with the hypomethylated/upregulated genes.

**Figure 6 ijms-23-08170-f006:**
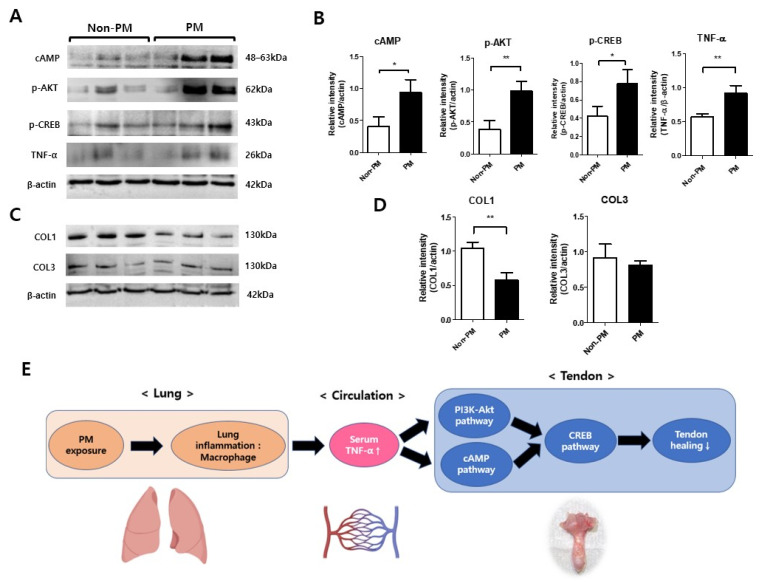
(**A**) Representative blots and (**B**) relative band intensities of cyclic adenosine monophosphate (cAMP), phosphorylated protein kinase B (p-Akt), phosphorylated cAMP response element-binding protein (p-CREB), and tumor necrosis factor-alpha (TNF-α). (**C**) Representative blots and (**D**) relative band intensities of collagen type I (COL1) and COL3 in the control (non-PM) and PM-exposed rats. The relative band intensity was normalized to β-actin. * *p* < 0.05, ** *p* < 0.01. (**E**) Proposed schematic diagram showing the action mechanism of PM exposure on tendon healing.

**Figure 7 ijms-23-08170-f007:**
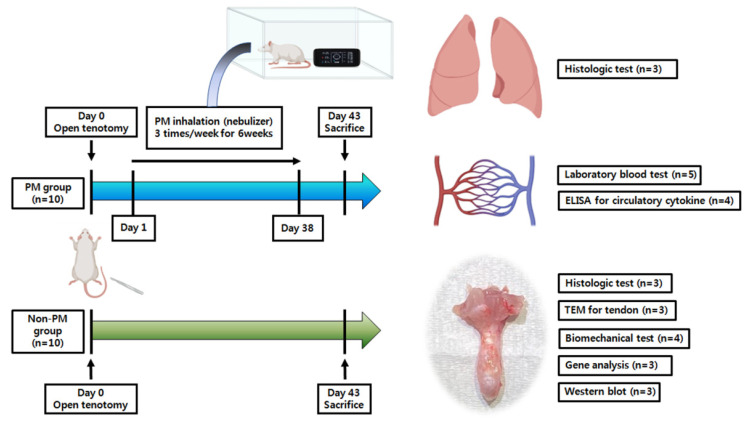
Overview of the experimental protocol. ELISA, enzyme-linked immunosorbent assay.

**Figure 8 ijms-23-08170-f008:**
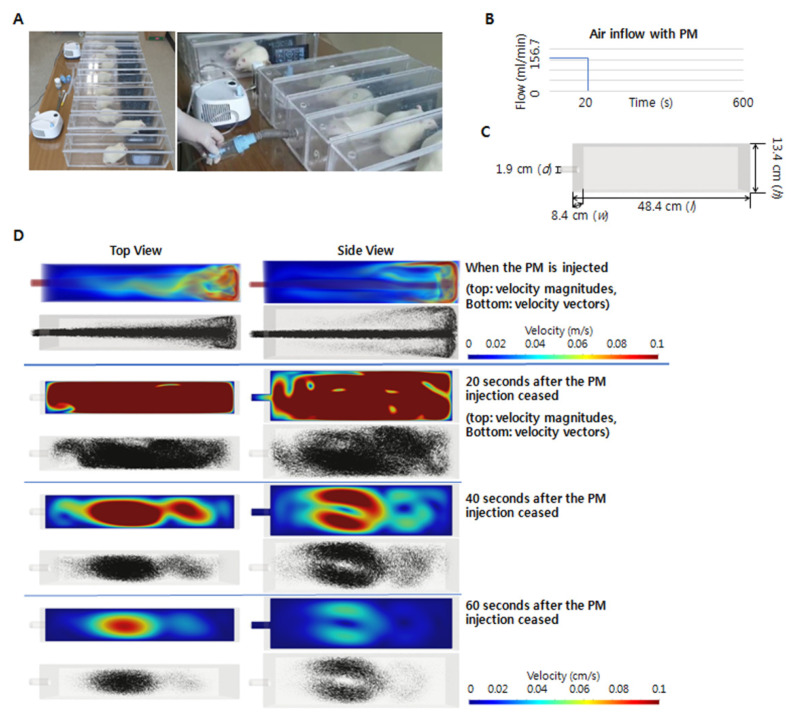
Methods followed for the PM exposure experiments. (**A**) Gross images of PM exposure using closed chambers and nebulizer systems. (**B**) Air inflow rate of the PM sample. (**C**) Dimensions of the closed chambers. (**D**) Simulations of the PM flow in the closed chamber 0, 20, 40, and 60 s after PM injection. d, diameter; w, width; l, length; h, height.

## Data Availability

Publicly available datasets were analyzed in this study. These data can be found here: GEO submission number GSE200709 and GSE200463.

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
