# Peer review of "Effects and Mechanism of Particulate Matter on Tendon Healing Based on Integrated Analysis of DNA Methylation and RNA Sequencing Data in a Rat Model"

_ijms, 2022, doi:10.3390/ijms23158170_

Round 1

Reviewer 1 Report

The manuscript of Lee “Effects and Mechanism of Particulate Matter on Tendon Healing based on Integrated Analysis of DNA Methylation and RNA Sequencing Data in a Rat Model” is devoted to the influence of particulate matter on tendon healing. The authors hypothesised that particulate matter could affect not only lungs and other known organs, but also tendons. No experimental study has been conducted to determine the relationship between tendon healing and particulate matter exposure.

The authors performed multi-pronged histological, ultrastructural, morphological and biomechanical investigations using well-established methods to reveal the effects of particulate matter in an Achilles tendon rupture rat model. Additionally, they performed the combined analysis of DNA methylation and RNA sequencing  data to elucidate the mechanism underlying the particulate matter -induced effects on tendon healing.

In general, the manuscript is well written and clearly describes the results. Method section clearly describes all the experimental procedures.

Discussion section is well written and mostly allows to compare the results obtained by the authors of the manuscript to the results of the other authors.

The main limitation of the work is the fact that the authors analyse already injured tendons and estimate the influence of particulate matter exposure indeed on the model of Achilles tendon rupture. Having in mind the general idea of contribution of particulate matter to human health issues it would be important to estimate the effects of particulate matters on the models without injuries. This point should be mentioned in the discussion section

Minor comment:

The authors write in the Discussion section

“these data suggest that PM exposure stimulated the TNF-α expression in the Achilles tendon via CREB pathway activation, subsequently inhibiting the COL1 expression and thereby inducing detrimental effects on tendon healing”

The study lacks any functional assays that would allow this conclusion; the data are purely descriptive in sense of functional relationships between pathways studied. This sentence should be omitted in the final version of the manuscript as not corresponding to the results presented.

Reviewer 2 Report

The authors have presented a thorough study of the impact the PM has on tendon healing using animal model. The study is experimentally sound, novel and provides useful insights for the field. I congratulate the authors for their comprehensive work.

Overall recommendation: Accept with minor changes

Please add/change the following the following:

1.     Please increase the size of the titles (total score, cell morphology etc)of the graphs in figure 2 B-G

2.     For lines 282-283, please add another line indicating the clinical significance of ‘decreased collagen diameter’ to further support your findings.

3.     Please re-title section 5 as Limitations and future directions and add a paragraph (current lines 340-346 as limitations) and another on the future directions.

4.     Please add section 6 as conclusions and expand on it (at least 7-8 lines). Iterate how this should be taken forward for applications in humans as well.
